# Optimal Design of Angular Displacement Sensor with Shared Magnetic Field Based on the Magnetic Equivalent Loop Method

**DOI:** 10.3390/s19092207

**Published:** 2019-05-13

**Authors:** Pinggui Luo, Qifu Tang, Huan Jing

**Affiliations:** Engineering Research Center of Mechanical Testing Technology and Equipment, Ministry of Education, Chongqing University of Technology, Chongqing 400054, China; tqf@cqut.edu.cn (Q.T.); jinghuan.viewhappy@2017.cqut.edu.cn (H.J.)

**Keywords:** angular displacement sensor, magnetic equivalent loop method (MELM), optimal design

## Abstract

Angular displacement sensor with shared magnetic field has strong environmental adaptability and high measurement accuracy. However, its 3-D structure is multi-pole double-layer structure, using time stepping finite element method (TSFEM) to optimize the structure is time-consuming and uneconomical. Therefore, a magnetic equivalent loop method (MELM) is proposed to simplify the optimal design of sensors. By reasonably setting the node position, the mechanical structure parameters, winding coefficients and input voltage of the sensor are integrated into a mathematical model to calculate of the induced voltage. The calculation results are compared with the simulation results, and a sensor prototype is made to test the optimized effect of the MELM.

## 1. Introduction

Angular displacement measurement widely exists in aircraft, ships and automobiles. It is the key technology that affects the performance and safety of these industrial products [1,2,3,4,5]. At present, there are grating sensors, capacitive sensors, resolvers and inductosyns for measuring angular displacement [6,7,8,9,10]. Grating sensors and capacitive sensors have high accuracy, but they are not suitable for working in harsh environment. The output signal of resolvers is strong, but the measurement accuracy is low. Inductosyns has high precision, but its output signal is weak, and subsequent circuit processing is difficult. Time grating angular displacement sensor as a kind of inductive angular displacement sensor, considers the advantages of resolvers and inductosyns [11,12]. It performs well under unfavorable conditions such as oil, water, dust, vibration, and temperature change, and is cheaper than most sensors with the same accuracy [13,14,15]. Angular displacement sensor with shared magnetic field is improved based on time grating angular displacement sensor. Two time grating angular displacement sensors are coaxially assembled, and two outer stators are merged into one stator. By applying excitation AC signals to the stator winding, the two inner rotors receive the same excitation magnetic field. This improved sensor is called the angular displacement sensor with shared magnetic field. It can output absolute angular position and realize self-correction [16].

In order to improve the measurement accuracy, increase the tooth number of stator and rotor is a main option. However, which will pose a severe challenge to optimize the sensor structure by time stepping finite element method (TSFEM). Since the double layer structure, the amount of simulation data is twice as much as before. Therefore, a new simplified method is urgently needed to optimize the structure. A magnetic equivalent circuit method (MELM) proposed in this paper greatly reduces the time of structural optimization, and the optimization results have been verified by simulation and experiment, which has a good application prospect. It should be pointed out that this paper is an extension of [17]. Compared with [17], this paper optimizes sensors with different structures and principles, better measurement accuracy and stronger desire for simplified optimization. Moreover, [17] do not give a general derivation formula of winding permeability, so that the solution of Induced voltage behind is actually problematic. This paper solves this problem and gives a complete mathematical model, which can be applied to other magnetic field sensors.

## 2. The Studied Sensors

### 2.1. Basic Structure of Sensor

The sensor consists of four parts: stator, rotor, excitation coil, and inducted coil, as shown in Figure 1.

Two sets of excitation coils are wound in the stator teeth, and the spatial phase difference is 90° in the circumferential direction. The specific turns distribution is as follows:(1)$$N_{a}=\frac{1}{2}N_{max}\sin P_{w1}\left[\frac{2\pi }{Z}\left(i-1\right)\right]+\frac{1}{2}N_{max}\sin P_{w2}\left[\frac{2\pi }{Z}\left(i-1\right)\right],$$
(2)$$N_{b}=\frac{1}{2}N_{max}\cos P_{w1}\left[\frac{2\pi }{Z}\left(i-1\right)\right]+\frac{1}{2}N_{max}\cos P_{w2}\left[\frac{2\pi }{Z}\left(i-1\right)\right],$$

$P_{w1}$ and $P_{w2}$ Pw2are the pole-pairs of two inner rotors, respectively, and Z is the number of teeth of stator.

Two sets of inducted coils are wound in two rotor teeth on equal turns to produce the maximum Induced voltage. The pole-pairs of two rotors are mutually prime. The end faces of stator teeth and rotor teeth are designed as T teeth to reduce magnetic leakage.

### 2.2. Measurement Principle of Sensor

The sensor is mainly designed according to the orthogonal law of the magnetic field in time domain and space domain, and the relationship between the measured angular displacement and the inducted voltage is established, as shown in Equation (3):(3)$$e\left(t,\theta \right)=U_{m}\sin\left(wt\right)\cos\left(\frac{2\pi }{P_{w}}\theta \right)+U_{m}\cos\left(wt\right)\sin\left(\frac{2\pi }{P_{w}}\theta \right)=U_{m}\sin\left(wt+\frac{2\pi }{P_{w}}\theta \right),$$
when w is the frequency of alternating current (AC) in the excitation coil, θ is the measured angular displacement, and $U_{m}$ is the amplitude of e(t,θ). According to Equation (3), the amplitude of traveling wave is determined by the excitation voltage and structure of sensor, and the phase varies with θ. Therefore, angular displacement can be obtained from a phase comparison between the traveling wave and the reference signal [18].

## 3. Magnetic Equivalent Loop Method

### 3.1. Mathematical Model

Firstly, it is necessary to simplify the structure of stator and rotor as much as possible while satisfying the basic shape, the simplified structure and size symbols are shown in Figure 2. Secondly, the stator and rotor are decomposed into sufficiently small parts, and reasonably selecting the node position in these parts, then the magnetic flux through the node remains constant. The magnetic potential of each node and the permeability between the nodes of the sensor are shown in Figure 3. Finally, before establishing the mathematical model, the following assumptions are made:
(1)Magnetic flux leakage between the nodes is neglected.(2)The influence of the excitation magnetic field diffused between the two inner rotors on the two rotors is neglected.

Firstly, assuming that $l_{0}$ is the axial length of the stator, $l_{X}$ and $l_{Y}$ are the axial lengths of two rotors (X and Y) respectively. For general expression, $l_{1}$ is the axial length of the rotor.

The magnetic flux at the yoke node of the jth rotor tooth can be expressed as:(4)$$\left(A_{j}^{ry}-A_{j-1}^{ry}\right)P_{j}^{ry}+\left(A_{j}^{ry}-A_{j+1}^{ry}\right)P_{j+1}^{ry}+\left(A_{j}^{ry}-A_{j}^{r}\right)P_{j}^{rr}=0,$$

Equation (4) can be expressed in matrix form:(5)$$A^{ry}P_{1}-A^{r}P_{2}=0,$$
where $A^{ry}={\left[A_{1}^{ry}\;A_{2}^{ry}\cdots A_{j}^{ry}\cdots A_{2P_{w}}^{ry}\right]}^{T}$ and $A^{r}={\left[A_{1}^{r}\;A_{2}^{r}\cdots A_{j}^{r}\cdots A_{2P_{w}}^{r}\right]}^{T}$.

The expression of magnetic flux and its matrix form at the middle node of the jth rotor tooth are as follows:(6)$$\left(A_{j}^{r}-A_{j-1}^{r}\right)P_{j}^{r}+\left(A_{j}^{r}-A_{j+1}^{r}\right)P_{j+1}^{r}+\left(A_{j}^{r}-A_{j}^{ry}\right)P_{j}^{rr}+\left(A_{j}^{r}-A_{j}^{rt}\right)P_{j}^{rt}=0,$$
(7)$$A^{r}P_{3}-A^{ry}P_{2}-A^{rt}P_{4}=0,$$

At the top node of the jth rotor tooth:(8)$$\left(A_{j}^{rt}-A_{j}^{r}\right)P_{j}^{rt}+\left(A_{j}^{rt}-A_{i-1}^{st}\right)P_{i-1,j}^{rs}+\left(A_{j}^{rt}-A_{i}^{st}\right)P_{i,j}^{rs}+\cdots =0,$$
(9)$$A^{rt}P_{5}-A^{r}P_{4}-A^{st}P_{6}=0,$$

At the top node of the ith stator tooth:(10)$$\frac{l_{1}\left(A_{i}^{st}-F_{i}^{M}-A_{j}^{s}\right)P_{i}^{M}}{l_{0}}+\left(A_{i}^{st}-A_{j}^{rt}\right)P_{i,j}^{rs}+\left(A_{i}^{st}-A_{j+1}^{rt}\right)P_{i,j+1}^{rs}+\cdots =0,$$
where $F_{i}^{M}=N_{i}^{M}I^{M}$.

(11)$$A^{st}P_{7}-A^{rt}P_{8}-\frac{l_{1}A^{s}P_{9}}{l_{0}}=\frac{l_{1}F^{M}P_{9}}{l_{0}},$$

At the middle node of the ith stator tooth:(12)$$\left(A_{i}^{s}+F_{i}^{M}-A_{i}^{st}\right)P_{i}^{M}+\left(A_{i}^{s}-A_{i-1}^{s}\right)P_{i}^{s}+\left(A_{i}^{s}-A_{i+1}^{s}\right)P_{i+1}^{s}+\left(A_{i}^{s}-F_{i}^{N}-A_{i}^{sy}\right)P_{i}^{N}=0,$$
where $F_{i}^{N}=N_{i}^{N}I^{N}$.

(13)$$A^{s}P_{10}-A^{st}P_{9}-A^{sy}P_{11}=-F^{M}P_{9}+F^{N}P_{11},$$

At the yoke node of the ith stator tooth:(14)$$\left(A_{i}^{sy}-A_{i-1}^{sy}\right)P_{i}^{sy}+\left(A_{i}^{sy}-A_{i+1}^{sy}\right)P_{i+1}^{sy}+\left(A_{i}^{sy}+F_{i}^{N}-A_{i}^{s}\right)P_{i}^{N}=0,$$
(15)$$A^{sy}P_{12}-A^{s}P_{11}=-F^{N}P_{11},$$

Equations (5), (7), (9), (11), (13), and (15) are combined as:
$$\left[\begin{array}{c} A^{ry}\\ A^{r}\\ A^{rt}\\ A^{st}\\ A^{s}\\ A^{sy} \end{array}]\times [\begin{array}{cccccc} P_{1} & -P_{2} & 0 & 0 & 0 & 0\\ -P_{2} & P_{3} & -P_{4} & 0 & 0 & 0\\ 0 & -P_{4} & P_{5} & -P_{6} & 0 & 0\\ 0 & 0 & -P_{8} & -P_{7} & \frac{l_{1}P_{9}}{l_{0}} & 0\\ 0 & 0 & 0 & -P_{9} & P_{10} & -P_{11}\\ 0 & 0 & 0 & 0 & -P_{11} & P_{12} \end{array}]=[\begin{array}{c} 0\\ 0\\ 0\\ \frac{l_{1}F^{M}P_{9}}{l_{0}}\\ -F^{M}P_{9}+F^{N}P_{11}\\ -F^{N}P_{11} \end{array}\right]$$
(16)AP=γ,

The structural parameters of the rotors X and Y are introduced into Equation (16) and the results are as follows:(17)$$A_{X}P_{X}=\gamma_{X},$$
(18)$$A_{Y}P_{Y}=\gamma_{Y},$$

### 3.2. Calculation of Permeability

Firstly, assuming that the rotor teeth thicknesses of X and Y are $d_{X}$ and $d_{Y}$, respectively: (19)$$P_{j}^{ry}=\frac{u_{0}u_{r}l_{1}}{\theta_{1}}\ln\frac{r_{2}}{r_{1}},$$
where $u_{r}$ is the relative permeability.

As shown in Figure 4a, $P_{j}^{rr}$ can be seen as a series connection between the permeability $P_{j}^{rr1}$ at the yoke of rotor tooth and the permeability $P_{j}^{rr2}$ at the Middle-lower part of rotor tooth:(20)$$\frac{1}{P_{j}^{rr}}=\frac{1}{P_{j}^{rr1}}+\frac{1}{P_{j}^{rr2}},$$
(21)$$P_{j}^{rr1}=\frac{u_{0}u_{r}l_{1}\theta_{1}\frac{r_{2}+r_{1}}{2}}{\frac{r_{2}-r_{1}}{2}},$$
(22)$$P_{j}^{rr2}=\frac{u_{0}u_{r}l_{1}d_{1}}{\frac{r_{3}-r_{2}}{2}},$$

As shown in Figure 4b, $P_{j}^{r}$ is connected in parallel by three parts: permeability $P_{j}^{r1}$ of inducted winding, permeability $P_{j}^{r2}$ of rotor tooth groove and permeability $P_{j}^{r3}$ of rotor tooth groove end:(23)$$P_{j}^{r}=P_{j}^{r1}+P_{j}^{r2}+P_{j}^{r3},$$
(24)$$P_{j}^{r1}=\frac{u_{0}l_{1}d_{1}}{d_{ind}},$$
where $d_{ind}$ is the winding length of inducted winding:(25)$$P_{j}^{r2}=\frac{u_{0}l_{1}}{\theta_{1}}\ln\frac{r_{3}}{r_{2}},$$
(26)$$P_{j}^{r3}=\frac{u_{0}l_{1}}{\theta_{3}}\ln\frac{r_{4}}{r_{3}},$$

In Figure 4c, $P_{j}^{rt}$ can be seen as a series connection of permeability $P_{j}^{rt1}$ at the middle-upper part of rotor tooth and permeability $P_{j}^{rt2}$ at the top of rotor tooth:(27)$$\frac{1}{P_{j}^{rt}}=\frac{1}{P_{j}^{rt1}}+\frac{1}{P_{j}^{rt2}},$$
(28)$$P_{j}^{rt1}=\frac{u_{0}u_{r}l_{1}\theta_{2}\frac{r_{3}+r_{4}}{2}}{\frac{r_{4}-r_{3}}{2}},$$
(29)$$P_{j}^{rt2}=\frac{u_{0}u_{r}l_{1}d_{1}}{\frac{r_{3}-r_{2}}{2}},$$

Air gap permeability $P_{i,j}^{rs}$ can be expressed as:(30)$$P_{i,j}^{rs}=\frac{u_{0}s_{i,j}}{d_{rs}},$$
where $d_{rs}$ is the length of air gap, $s_{i,j}$ is the contact area between the ith stator tooth and jth rotor tooth:(31)$$\frac{1}{P_{i}^{M}}=\frac{1}{P_{i}^{M1}}+\frac{1}{P_{i}^{M2}},$$
(32)$$P_{i}^{M1}=\frac{u_{0}u_{r}l_{0}\theta_{5}\frac{r_{5}+r_{6}}{2}}{\frac{r_{6}-r_{5}}{2}},$$
(33)$$P_{i}^{M2}=\frac{u_{0}u_{r}l_{0}d_{2}}{\frac{r_{7}-r_{6}}{2}},$$
(34)$$P_{i}^{s}=P_{i}^{s1}+P_{i}^{s2}+P_{i}^{s3}+P_{i}^{s4},$$
(35)$$P_{i}^{s1}=\frac{u_{0}l_{0}d_{2}}{d_{M}},$$
(36)$$P_{i}^{s2}=\frac{u_{0}l_{0}d_{2}}{d_{N}},$$
where $d_{M}$ is the winding length of excitation winding (M), $d_{N}$ is the winding length of excitation winding (N):(37)$$P_{i}^{s3}=\frac{u_{0}l_{0}}{\theta_{4}}\ln\frac{r_{7}}{r_{6}},$$
(38)$$P_{i}^{s4}=\frac{u_{0}l_{0}}{\theta_{6}}\ln\frac{r_{6}}{r_{5}},$$
(39)$$\frac{1}{P_{i}^{N}}=\frac{1}{P_{i}^{N1}}+\frac{1}{P_{i}^{N2}},$$
(40)$$P_{i}^{N1}=\frac{u_{0}u_{r}l_{0}\theta_{4}\frac{r_{7}+r_{8}}{2}}{\frac{r_{8}-r_{7}}{2}},$$
(41)$$P_{i}^{N2}=\frac{u_{0}u_{r}l_{0}d_{2}}{\frac{r_{7}-r_{6}}{2}},$$
(42)$$P_{i}^{N2}=\frac{u_{0}u_{r}l_{0}d_{2}}{\frac{r_{7}-r_{6}}{2}},$$

### 3.3. Calculation of Induced Voltage

At every time, the $A_{j}^{r}$ on each rotor tooth is calculated by calculating the magnetic permeance of each node and the winding magnetic potential generated by two excitation coils. Then the magnetic flux $\Phi_{1}$ of inducted coil can be calculated:(43)$$\Phi_{1}=\sum_{j=1}^{2P_{w}}P_{j}^{r1}\cdot A_{j}^{r},$$

Finally, the inducted voltage of inducted coil is calculated:(44)$$U_{out}=\frac{d\Phi_{1}}{dt},$$

Meanwhile, it should be noted that the inducted magnetic potential is also generated in the excitation coil. The two sets of excitation currents change with the variance of the magnetic potential $A^{s}$ at the middle of the stator teeth. It can be expressed as:(45)$$\{\begin{array}{c} U_{in1}=\frac{d\Phi_{2}}{dt}+I_{M}R_{M}\\ U_{in2}=\frac{d\Phi_{3}}{dt}+I_{N}R_{N} \end{array},$$
where $\{\begin{array}{c} \Phi_{2}=\sum_{j=1}^{2P_{w}}P_{i}^{s1}\cdot A^{s}\\ \Phi_{3}=\sum_{j=1}^{2P_{w}}P_{i}^{s2}\cdot A^{s} \end{array}$.

The whole calculation process of MELM can be expressed as:
Through (19) to (29), the permeability $P^{ry}$, $P^{r}$, $P^{rr}$, and $P^{rt}$ of each node of the rotor X and Y are calculated, respectively.Through Equations (30)–(42), calculating the air gap permeability $P^{rs}$ and the permeability $P^{M}$, $P^{s}$, $P^{N}$ and $P^{sy}$ of the stator nodes.By Equations (17) or (18), the magnetic potential $A^{s}$ at the middle of stator teeth is calculated.Updating excitation current $I_{M}$ and $I_{N}$ by Equations (45)Again, by Equations (18) and (19), the magnetic potential $A^{r}$ at the middle of the teeth of rotor X and Y is calculated, respectively.Through respectively calculating the induced voltage $U_{Xout}$ and $U_{Yout}$ of rotor X and Y by Equations (43) and (44).When rotating, the air gap permeability $P^{rs}$ changes, and the induced voltage is recalculated by repeating processes 3–6.

### 3.4. Simulation Verification

In order to judge whether the magnetic equivalent loop method is correct, a small number of simulations are needed to verify it. To reduce the amount of simulation, the pole-pairs of selected rotor is less than 30, and the number of stator teeth is limited to less than 50. As is well known, the time and accuracy of TSFEM is closely related to the size of mesh generation. According to the past simulation experience, the mesh length of stator and rotor is set to 3 mm and the mesh length of coils is set to 1 mm in this simulation. The mesh diagram of the simulation sensor is shown in Figure 5, besides, Table 1 is specific structural parameters, and electrical parameters are shown in Table 2.

In FEM simulation and MELM calculation, five high frequency excitation period and 20 points in one period are considered. Figure 6 shows the waveforms of inducted voltage obtained by FEM simulation and calculated by MELM, respectively, when the inner rotor X and Y are excited by two phases.

Assuming that the inducted voltage values of the rotor are $V_{t=k}^{TSFEM}$ and $V_{t=k}^{MELM}$ at each time through FEM and MELM respectively, the difference between the two inducted voltages can be expressed as:(46)$$\Delta =\sum_{k=1}^{100}\Delta_{k}=\sum_{k=1}^{100}\left|\frac{V_{t=k}^{TSFEM}-V_{t=k}^{MELM}}{100\cdot V_{t=k}^{TSFEM}}\right|,$$

The values of $\Delta_{k}$ is shown in Figure 7.

The Δ value of rotor X is 1.57%, and that Δ value of rotor Y is 2.43%. Both Δ values are less than 3%. It takes 24 h to simulate with ANSYS Maxwell software (ANSYS, Inc., USA), while MELM on MATLAB software (The MathWorks, Inc, America) takes less than 5 min to compute. Therefore, it is feasible and time-saving to calculate the inducted voltage of the sensor by MELM.

## 4. Optimal Design with MELM

The optimization of angular displacement sensor should focus on the structure of stator. Since the stator provides the shared excitation magnetic field to the inner two-layer rotor, the structure of the stator directly determines the quality of the excitation magnetic field. Adapting control variable method to optimize the stator structure and ensure that the structural dimensions of the two inner rotors remain unchanged. The optimum contents of stator structure are as follows: trapezoidal groove clearance $\theta_{6}$, trapezoidal groove height $r_{6}-r_{5}$, stator teeth width $d_{2}$, and air gap length $d_{rs}$. Firstly, in order to compare with the past experimental results of [16], and the number of rotor X and rotor Y must be mutual prime to realize self-correction. Thus, the number of stator teeth is 144, the number of rotor X teeth is 50, the number of rotor Y teeth is 128. The specific structural dimensions of the preparative optimal sensor are shown in Table 3. Meanwhile, two criteria for judging the optimization results should be defined. The first is the average value of the induced voltage (VAA) under two-phase excitation, and the second is the discrete coefficient of the induced voltage amplitude change (VADC) under two-phase excitation. Minimum VADC must be considered as a priority when meeting the appropriate VAA. It is noteworthy that the VAA and VADC of the induced voltage here are the sum of the VAA and VADC of the induced voltage of the two inner rotors.

### 4.1. Trapezoidal Groove Clearance $\theta_{6}$

The trapezoidal groove clearance $\theta_{6}$ mainly influences the convenience of winding process. In Figure 8, The VAA increases with the increase of the trapezoidal groove clearance $\theta_{6}$, and the VADC change decreases firstly and then increases with the increase of the trapezoidal groove clearance $\theta_{6}$ under the two-phase excitation. when $\theta_{6}$ equals 0.3${}^{{}^{\circ}}$, VADC is the smallest and which can satisfy the width of winding into the stator teeth groove. $\theta_{6}=0.3^{{}^{\circ}}$ is the best choice.

### 4.2. Trapezoidal Groove Height $r_{6}-r_{5}$

Trapezoidal groove height $r_{6}-r_{5}$ influences the mechanical strength of stator teeth. Additionally, the increase of trapezoidal groove height reduces the number of excitation coils turns. From Figure 9, the VAA decreases with the increase of trapezoidal groove height $r_{6}-r_{5}$. When $r_{6}-r_{5}=$ 0.5 mm, VADC is the smallest, and which can also meet the basic mechanical strength requirements of rotation. $r_{6}-r_{5}=$ 0.5 mm can be used as the best option.

### 4.3. Stator Teeth Width $d_{2}$

Figure 10 shows that the VAA increases slowly with the increase of stator tooth width $d_{2}$ under two-phase excitation, and the VADC also increases slowly with the increase of stator tooth width $d_{2}$. Considering the mechanical strength, mechanical deformation and other factors, the stator tooth width $d_{2}$ cannot be too small. Considering comprehensively, the stator tooth width $d_{2}=$ 2 mm is chosen.

### 4.4. Air-Gap Length $d_{rs}$

As shown in Figure 11, the VAA decreases rapidly with the increase of air-gap length $d_{rs}$, and the VADC also increases rapidly with the increase of air-gap length $d_{rs}$ under two-phase excitation. Because of the high magnetoresistance of air, the conduction of voltage decreases rapidly. Therefore, it is necessary to find a balance point between VAA and VADC, and mainly focus on VAA, choose the air-gap length $d_{rs}=$ 0.3 mm.

## 5. Experimental Verification

According to the results of MELM optimal design, the sensor prototype is made with the dimensions of $\theta_{6}=$0.3${}^{{}^{\circ}}$, $r_{6}-r_{5}=$ 0.5 mm, $d_{2}=$ 2 mm, and $d_{rs}=$ 0.3 mm. Figure 12 is the experimental system. The marble frame is a platform for installing the angular displacement sensor, grating encoder, rotary workbench, and motor. The angular displacement sensor and the grating encoder are coaxial, and they are driven by the rotary workbench controlled by the motor servo system. The sensor error are obtained by using the HEIDENHAIN RON 886 grating encoder (HEIDENHAIN, Ltd, Germany) with system accuracy of ±1 arcsec. An industrial computer is equipped for the motor servo system, and relevant circuits and program software are designed for data acquisition. The position data of the sensor and grating encoder are collected at the same time and transmitted to the notebook computer. In order to make the winding more convenient and faster, copper wire wrapped in plastic sheet is used in the winding, which reduces the time cost by 50% compared with the previous process. Although the actual inducted voltage has been reduced, it is still over 100 mV and meets the basic processing circuit requirements.

The sensor prototype is sampled at the excitation voltage of 5 kHz and 4 V, and the rotational speed of motor is 0.36r/min, 22r/min. The purpose of setting different rotational speeds is to eliminate the influence of rotational speeds on errors. Additionally, at lower speed, more sampling points can be used as standard error values. The one period measurement error and the full range measurement errors of rotor X and Y are obtained, as shown in Figure 13, Figure 14, Figure 15 and Figure 16. In theory, the more teeth, the average effect of restraining machining and installation errors is greater and the more frequency errors in a cycle will be restrained [19]. From Figure 13 and Figure 14, the one period measurement error of rotor X and rotor Y is different, and it decreases with the increase of the number of rotor teeth. Meanwhile, compared Figure 13 and Figure 14 with Figure 15 and Figure 16, because of the Doppler effect [20], the higher the rotational speed, the greater the measurement error.

Compared with [16] without structural optimization, the error peak decreases by about 50%. The experimental results should be pointed out that the manufacturing error and installation error of the sensor should be excluded. In addition to manufacture and installation, this experiment uses the same winding, circuit, and signal processing program as [16].

## 6. Conclusions

In this paper, a magnetic equivalent loop method (MELM) is proposed to optimize the design of the sensor. The mechanical structure parameters, winding coefficient, and input voltage of the sensor are integrated into a mathematical model by reasonably setting the node position to complete the calculation of the output voltage. Comparing the calculated results with the simulation results, the difference is less than 3%. Additionally, compared with the time stepped finite element method (TSFEM), this method can greatly save the time of structural optimization. Finally, a prototype is made and tested to verify the optimization effect of MELM.

## Figures and Tables

**Figure 1 sensors-19-02207-f001:**
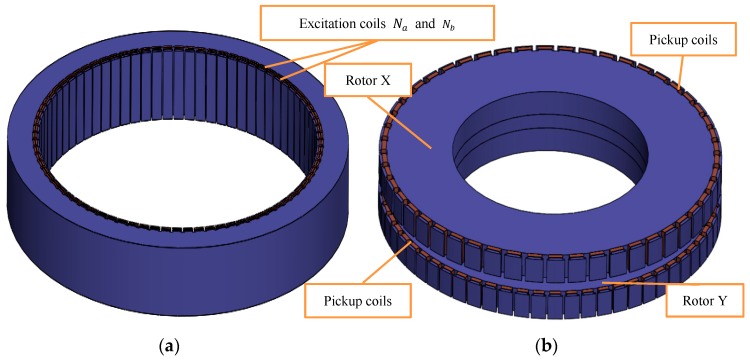
The structure of sensor: (**a**) Stator; and (**b**) rotor.

**Figure 2 sensors-19-02207-f002:**
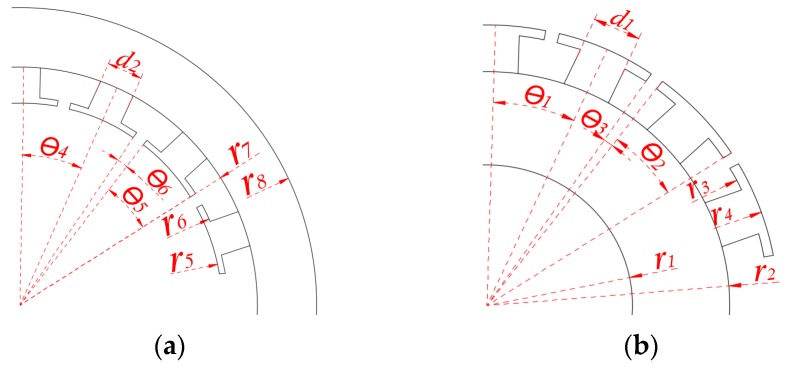
The simplified structure and size symbols: (**a**) Stator; and (**b**) rotor.

**Figure 3 sensors-19-02207-f003:**
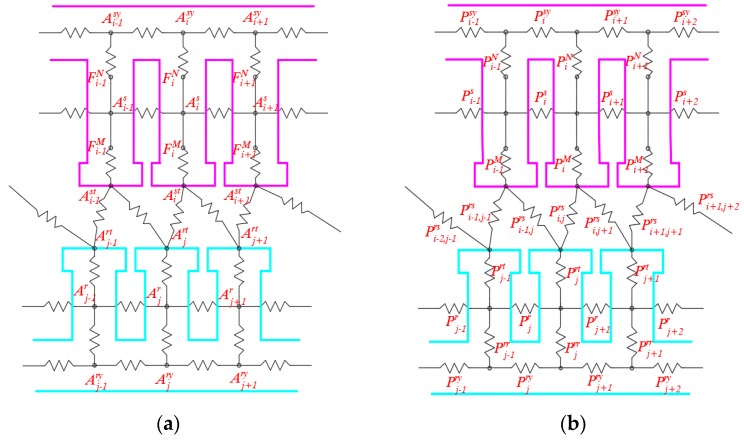
The magnetic equivalent loop method of sensor: (**a**) The magnetic potential of each node; and (**b**) the permeability between the nodes.

**Figure 4 sensors-19-02207-f004:**
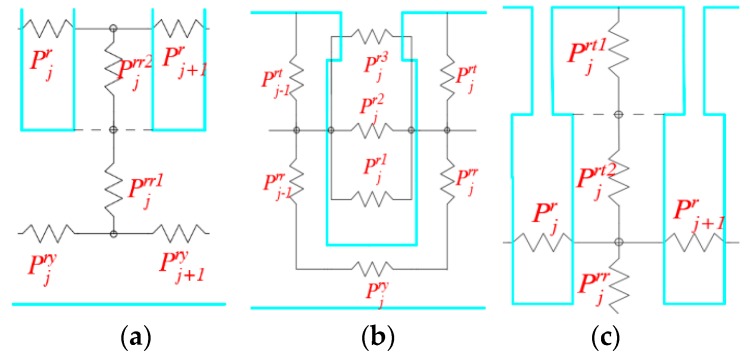
Decomposition diagram of permeability at each node of rotor tooth: (**a**) Decomposition diagram of $P_{j}^{rr}$; (**b**) decomposition diagram of $P_{j}^{r}$; and (**c**) decomposition diagram of $P_{j}^{rt}$.

**Figure 5 sensors-19-02207-f005:**
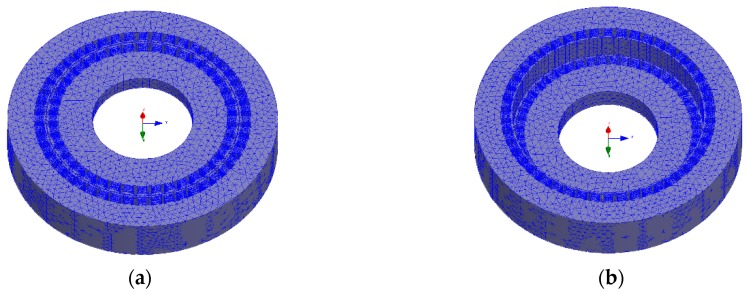
The mesh diagram of the simulation sensor: (**a**) Rotor X and stator; and (**b**) rotor Y and stator.

**Figure 6 sensors-19-02207-f006:**
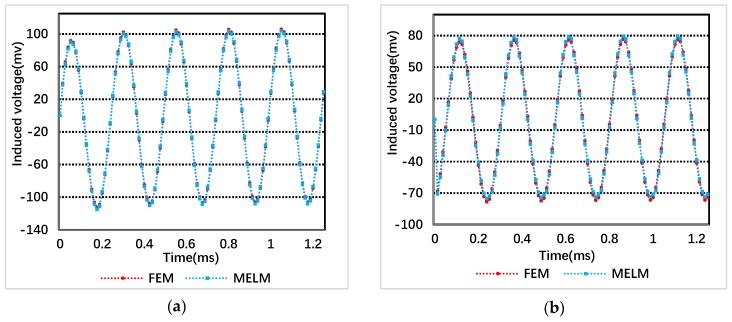
Induced voltage waveform of the inner rotor by FEM and MELM under two-phase Excitation: (**a**) Rotor X; and (**b**) rotor Y.

**Figure 7 sensors-19-02207-f007:**
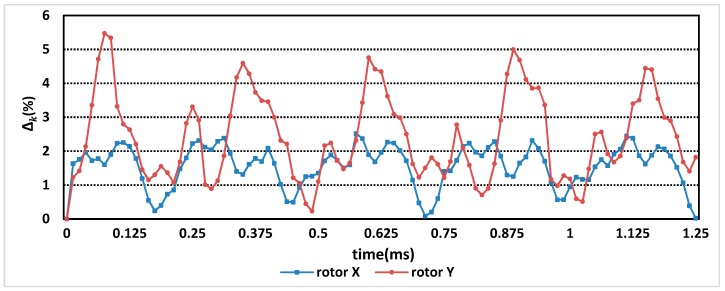
The values of $\Delta_{k}$ of rotor X and rotor Y.

**Figure 8 sensors-19-02207-f008:**
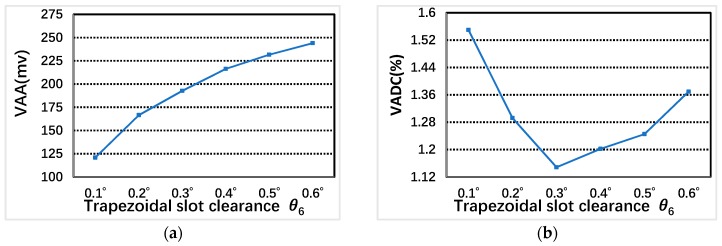
The relationship between VAA, VADC and trapezoidal groove clearance $\theta_{6}$: (**a**) VAA; and (**b**) VADC.

**Figure 9 sensors-19-02207-f009:**
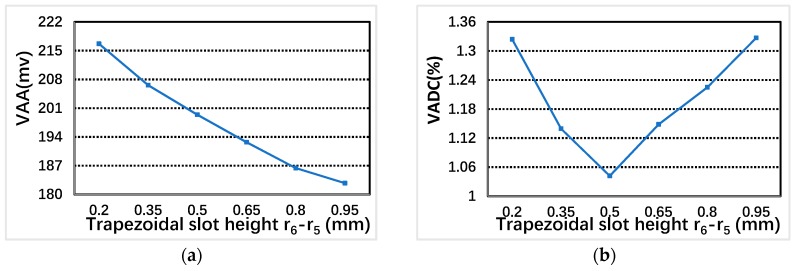
The relationship between VAA, VADC and trapezoidal groove height $r_{6}-r_{5}$: (**a**) VAA; and (**b**) VADC.

**Figure 10 sensors-19-02207-f010:**
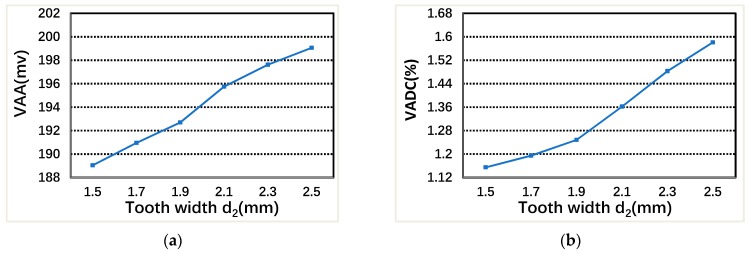
The relationship between VAA, VADC, and stator teeth width $d_{2}$: (**a**) VAA; and (**b**) VADC.

**Figure 11 sensors-19-02207-f011:**
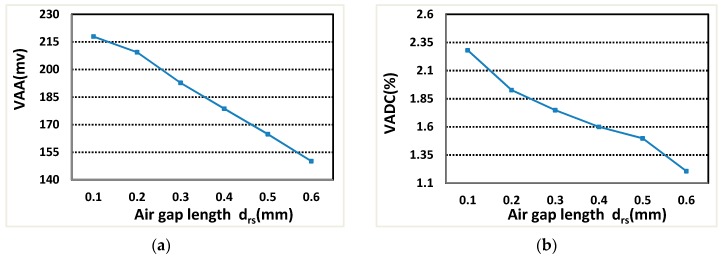
The relationship between VAA, VADC and air-gap length $d_{rs}$: (**a**) VAA; and (**b**) VADC.

**Figure 12 sensors-19-02207-f012:**
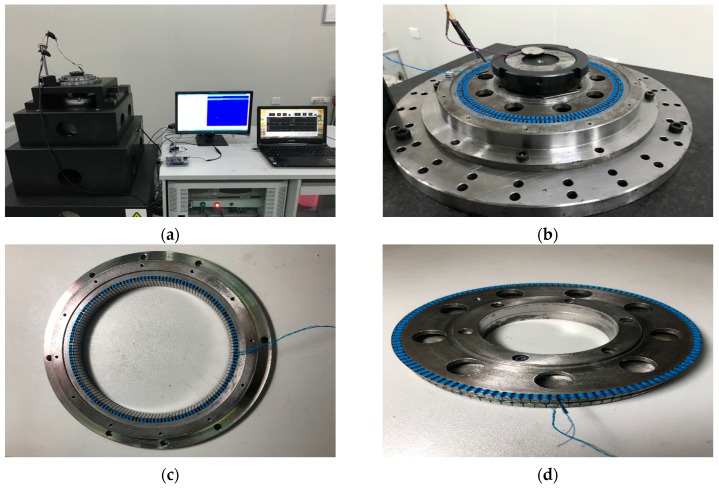
Experimental system: (**a**) Experimental table; and (**b**) sensor prototype; (**c**) stator; (**d**) rotor.

**Figure 13 sensors-19-02207-f013:**
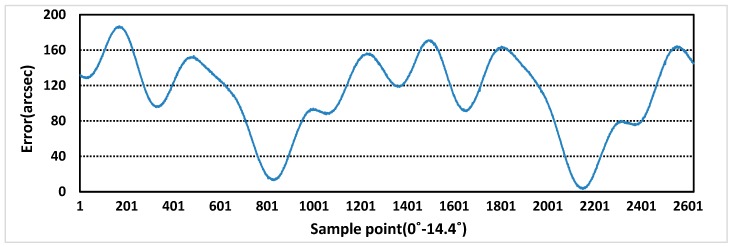
One period measurement error of rotor X at 0.36 r/min.

**Figure 14 sensors-19-02207-f014:**
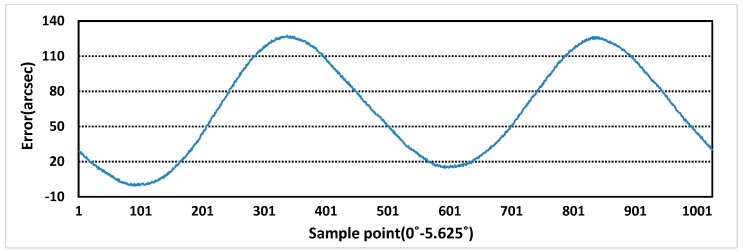
One period measurement error of rotor Y at 0.36 r/min.

**Figure 15 sensors-19-02207-f015:**
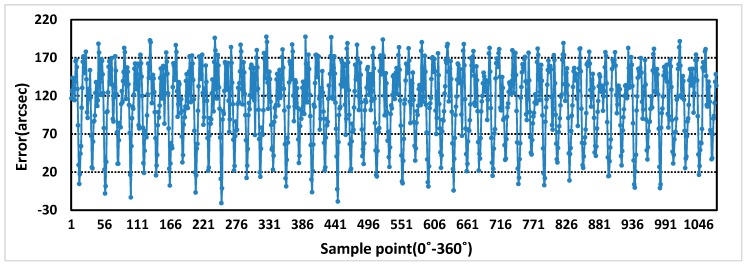
Full range measurement error of rotor X at 22 r/min.

**Figure 16 sensors-19-02207-f016:**
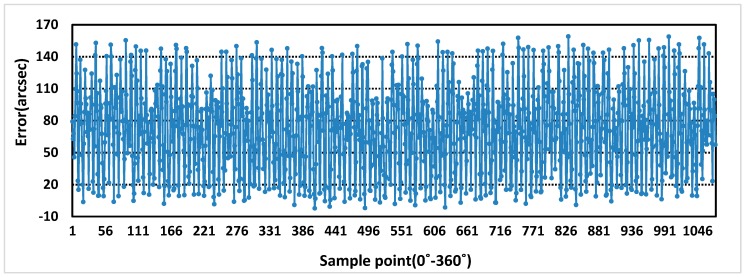
Full range measurement error of rotor Y at 22 r/min.

**Table 1 sensors-19-02207-t001:** Structural parameters of simulation.

Size Symbol	Structural Parameters
$r_{1}/r_{2}/r_{3}/r_{4}$(mm)	30/50/56/57
$r_{5}/r_{6}/r_{7}/r_{8}$(mm)	57.3/58.3/64.3/80
$\theta_{1}^{X}/\theta_{2}^{X}/\theta_{3}^{X}$	$9^{{}^{\circ}}/8^{{}^{\circ}}/1^{{}^{\circ}}$
$\theta_{1}^{Y}/\theta_{2}^{Y}/\theta_{3}^{Y}$	$7.2^{{}^{\circ}}/6.5^{{}^{\circ}}/0.7^{{}^{\circ}}$
$\theta_{4}/\theta_{5}/\theta_{6}$	$7.5^{{}^{\circ}}/7^{{}^{\circ}}/0.5^{{}^{\circ}}$
$l_{X}/l_{Y}/l_{0}$(mm)	10/13/30
$d_{X}/d_{Y}/d_{2}$(mm)	5/4/4.5

**Table 2 sensors-19-02207-t002:** Electrical parameters of simulation.

Electrical Symbols	Electrical Parameters
Excitation voltage frequency (kHz)	4
Excitation voltage amplitude (V)	5
Excitation coil resistance (Ω)	10
Inducted coil resistance (MΩ)	10
Excitation coil maximum turns	20
Inducted coil turns	1
Rotate speed (r/min)	20

**Table 3 sensors-19-02207-t003:** Structural parameters of the preparative optimal sensor.

Size Symbol	Structural Parameters
$r_{1}/r_{2}/r_{3}/r_{4}$(mm)	30/60/64/64.8
$r_{7}/r_{8}$(mm)	70/85
$\theta_{3}^{X}/\theta_{3}^{Y}$	$0.8^{{}^{\circ}}/0.3^{{}^{\circ}}$
$l_{X}/l_{Y}/l_{0}$(mm)	5/5/14
$d_{X}/d_{Y}$(mm)	5/2

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
