# Peer review of "Optimal Design of Angular Displacement Sensor with Shared Magnetic Field Based on the Magnetic Equivalent Loop Method"

_sensors, 2019, doi:10.3390/s19092207_

Round 1

Reviewer 1 Report

Dear Author,

I have read the manuscript “Optimal Design of Angular Displacement Sensor with Shared Magnetic Field Based on Magnetic Equivalent Loop Method.” The technique demonstrated is interesting and I’m impressed by the successful use of the a magnetic equivalent loop method (MELM) is proposed to simplify the optimal design of sensors . I recommend publication after minor revisions.

When you assume that the inductor voltage values of the rotor are ?? = ?????? and ?? = ????? at each time through FEM and MELM respectively, the difference between the two inducted voltages can be expressed by equation (46), which you conclude the percentage of 2.43% less than 3%. It was not very clear from the text the comparing the calculated results with the simulation results. 

Perhaps explaining with more details the process of acquisition of the theoretical data:

"It takes 24 hours to simulate with ANSYS Maxwell software on Inter Core i7-6700HQ, 8GB DDR3L and 1TB HD laptops, while MELM on Matlab software takes less than 5 minutes to compute. Therefore, it is feasible and time-saving to calculate the induction voltage of the sensor by MELM. "

Author Response

Dear Reviewer,

Thank you very much for your comments, Here are my modifications:

(1) In order to express the difference between the calculated results and simulation results more clearly, equation (46) is changed, adds a symbol Δk to represent differece values for each time interval and adds Figure 7 of Δk to conclude the percentage of 2.43% less than 3%.

(2) For explaining with more details the process of acquisition of the theoretical data, I mainly explains the mesh length of stator, rotor and coils in this simulation and adds the mesh diagram of the simulation sensor (Figure 5).

All the changes in this paper have been highlighted and underlined in red.

Again, I would like to express my gratitude.

Reviewer 2 Report

Review comments for Sensors - 494388:

The authors propose the magnetic equivalent loop method to optimize the sensor structures. In my opinion, this paper can be considered acceptance after major revisions according to the suggested comments.

(1) The symbol w shall be frequency in line 69.

(2) In lines 163~165, the symbol “MECM” is error.

(3) In Figures 11, 12, how about the rotation speed for the measurement? Not only the raw data, but also it shall be better to provide the values of standard error. Moreover, the comment regarding the different errors shall be described for the two rotators. Based on the optimal structures, please comment the statistical error depending on the rotational speed of rotator.

(4) In lines 167~168, does it mean that the TSFEM needs 24hrs to calculate the inducted voltage waveform?

(5) In Figure 1(b), the rotor X and Y shall be marked in the inner two-layer rotor.

(6) In lines 178~179, how to optimize the teeth numbers for the stator, rotor X, and rotor Y? It is better to provide some description for the design criteria.

(7) The values of VAA and VADC are the main factors for the repeatability and stability in the induced voltage waveforms. Which of the VAA and VADC parameters is the main influence factor on the phase measurement precision?

(8) During the review period, the cited references [16] are just accepted and not disclosed on line, it is difficult to compare the reference data with the submitted results. The authors shall to replace another related reference. By the way, the stability and resolution of the rotatory angular displacement sensor shall be quantitative. It is better to compare with another related research results. 

Author Response

Dear Reviewer,

Thank you very much for your comments, Your professional and detailed comments are the ladder of my progress. Here are my modifications:

(1)The symbol w has been changed as frequency.

(2) The symbol “MECM” has been changed as "MELM". 

(3) In original Figures 11, 12, the rotation speed for the measurement is 22r/min. I has gived the values of standard error at 0.36r/min in Figure 13 and 14. and the reason of the different errors of two rotators is the more teeth, the average effect of restraining machining and installation errors is greater and the more frequency errors in a cycle will be restrained, Based on the optimal structures, the statistical error depending on the rotational speed of rotator is caused by the Doppler effect.

(4) The time of using TSFEM to calculate the inducted voltage waveform is depending on the mesh generation. in this simulation, the mesh length of stator and rotor is set to 3 mm and the mesh length of coils is set to 1 mm. It takes 24 hours to simulate with ANSYS Maxwell software. I adds the mesh diagram of the simulation sensor (Figure 5).

(5) In Figure 1(b), Rotor X and Y, Pick coils and Excitation coils have marked in the stator and the inner two-layer rotor.

(6) As for the teeth numbers for the stator, rotor X, and rotor Y, I explains that in order to compare with the past experimental results of [16], and the number of rotor X and rotor Y must be mutual prime to realize self-correction.

(7) The VADC parameters is the main influence factor on the phase measurement precision. In this paper, I explains that Minimum VADC must be considered as a priority when meeting the appropriate VAA.

(8) The cited references [16] has been Early accessed, which has been retrieved in https://ieeexplore.ieee.org/document/8680022. So, [16] can be the compare as another related research results. 

All changes have been highlighted and underlined in red, and whose are equipped with serial number for you to check.

Once again, I want to express my gratitude to you.

Round 2

Reviewer 2 Report

The authors solved the concerns provided by the reviewers and the paper seems more suitable for acceptance after the corrections.